# Suggestions for Standardized Identifiers for Fatty Acyl Compounds in Genome Scale Metabolic Models and Their Application to the WormJam *Caenorhabditis elegans* Model

**DOI:** 10.3390/metabo10040130

**Published:** 2020-03-28

**Authors:** Michael Witting

**Affiliations:** 1Research Unit Analytical BioGeoChemistry, Helmholtz Zentrum München, Ingolstädter Landstraße 1, 85764 Neuherberg, Germany; michael.witting@helmholtz-muenchen.de; 2Analytical Food Chemistry, TU München, Maximus-von-Imhof-Forum 2, 85354 Freising, Germany

**Keywords:** Genome scale metabolic networks, *Caenorhabditis elegans*, standardization

## Abstract

Genome scale metabolic models (GSMs) are a representation of the current knowledge on the metabolism of a given organism or superorganism. They group metabolites, genes, enzymes and reactions together to form a mathematical model and representation that can be used to analyze metabolic networks in silico or used for analysis of omics data. Beside correct mass and charge balance, correct structural annotation of metabolites represents an important factor for analysis of these metabolic networks. However, several metabolites in different GSMs have no or only partial structural information associated with them. Here, a new systematic nomenclature for acyl-based metabolites such as fatty acids, acyl-carnitines, acyl-coenzymes A or acyl-carrier proteins is presented. This nomenclature enables one to encode structural details in the metabolite identifiers and improves human readability of reactions. As proof of principle, it was applied to the fatty acid biosynthesis and degradation in the *Caenorhabditis elegans* consensus model WormJam.

## 1. Introduction

Genome scale metabolic models (GSMs) are a representation of the current knowledge on the metabolism of a given organism or superorganism. They group metabolites, genes, enzymes and reactions together to form a mathematical model and representation that can be used to analyze metabolic networks in silico or used for analysis of omics data [1].

Typically, GSMs are reconstructed by applying a variety of protocols to resources such as genome annotations and existing pathway reconstructions [2]. Following rigorous manual curation, networks can be used for a range of purposes. An important determinant in the quality of a model is the accurate depiction of the individual entities present in the reaction network, e.g., metabolites and cofactors. The annotation of metabolites in particular can have a large impact on the model. Metabolites must be represented at the correct charge state and molecular formula to define mass and charge balanced reactions, required for correct predictions [3].

Standardization of the representation of these models is an important issue to enhance reuse of GSMs throughout the wider scientific community, facilitate comparison across different models and allow merging of different models into larger models (e.g., host–microbiome models).

While each metabolite has a unique chemical structure, which allows its unambiguous identification, GSMs often rely on simple identifiers and sum formulae, and only a few metabolic reconstructions have detailed curation of chemical structures associated with the model. Examining nearly 100 published GSMs, Ravikrishnan and Raman found that over 60% of these models were lacking standard metabolite identifiers such as KEGG IDs, Pubchem IDs and InChIs [4]. Currently, a large community effort towards standardization of GSMs has been started [5,6].

Several GSMs for the biomedical model organisms *Caenorhabditis elegans* have been described and recently merged into the WormJam consensus model [7,8,9,10]. This model aims to represent accurately the current knowledge on *C. elegans* metabolism to facilitate both modeling and metabolomics data analysis.

During work on the WormJam model, it was realized that lipid metabolism is a major flaw of both the constituent models and the merged model [10,11]. Several reactions are either only present as lumped or nested reactions or contain major mistakes. Furthermore, metabolite naming and naming of identifiers between the models are very different, which led to several duplicated reactions that could not be easily detected in an automatic manner. Unique identifiers of metabolites are used in the reactions, which must uniquely identify a metabolite within the reaction network of a GSM. The WormJam model and several other GSMs use the BiGG identifiers as common metabolite identifiers in the model reactions [12]. These identifiers are short and should be human readable. While for well-established and defined metabolites (e.g., amino acids) this is easily achievable, others often have cryptic names. For example, metabolites such as L-alanine are easily abbreviated as “ala__L”, whereas 12-hydroxyeicosatetraenoate is abbreviated as CE0347 in Recon3D for example. While being unique, this identifier is not human readable, which makes required human curation of metabolic pathways complicated. Manual curation of the *C. elegans* fatty acid biosynthesis and β-oxidation as part of lipid metabolism showed that identifiers of related metabolites are very heterogenous between the models, but also within BiGG. While BiGG and the MetaNetX namespace enable bridging between reactions and metabolites, the use of “non-standard” metabolite IDs complicates this process [13].

We found that many different identifiers exist for fatty acid and fatty acyl-based metabolites like acyl-Coenzymes A (acyl-CoAs) or acyl-carnitines. While identifiers within one class (e.g., fatty acids) are often in good agreement and comparable, a comparison between different classes is often difficult. This makes it difficult to achieve human readability. A particular example in BiGG is the metabolite anteiso C17:0 CoA (15-methyl-hexadecanoyl-CoA), which has the identifier “fa12coa” in the *Bacillus subtilis* model iYO844 [14]. This can be very misleading, since the identifier had no more relation with the actual chemical entity and could be mistaken as lauroyl-CoA. Another example from the same model is the fatty acid c15:0iso (13-methyl-tetradecanoate), which had the ID “fa3”. Such ambiguous naming needs to be overcome to allow better comparison between different models and facilitate further model development. Furthermore, ambiguous naming with no further detail of structural information is often used, e.g., “tetradecenoyl carnitine”, which refers to an acyl carnitine with 14 carbons and one double bond. However, since no exact structural details are given either in the ID or name, and there are no links to chemical databases, a definite structural identification of the molecular entity is not possible.

In this paper, a naming scheme for metabolites like fatty acids, acyl-CoAs and acyl-carnitines is proposed based on an established shorthand notation for lipids [15]. This naming scheme makes use of the defined building blocks, such as the number of carbons, double bonds, and functional groups and uses them to create unambiguous IDs. The naming scheme uses a common namespace to describe the acyl moiety for different covered classes (fatty acid, acyl-carrier proteins (ACPs), acyl-CoAs, acyl-carnitines and N-acyl-ethanolamides). Using this scheme, human curation is more easily achievable and comparison between different GSMs is facilitated by using common IDs. Since the scheme follows defined rules, it is human and machine readable. The proof-of-concept data from the WormJam model and BiGG have been analyzed on the validity of structural detail of their ID and metabolite names in the context of structural information content. New systematic IDs have been generated for metabolites for which definite structures have been concluded. Lastly, reactions related to fatty acid biosynthesis and degradation have been generated for the WormJam, and the new systematic IDs have been used to showcase the ease of use of this new system.

## 2. Materials and Methods

### 2.1. Caenorhabditis elegans Models

The SBtab version of the original models of iCel1273 and ElegCyc were used [16]. For the merged WormJam model, an updated, intermediate version of the previously published version was used [10]. This version from the date 01/08/2019 is available from GitHub (https://github.com/JakeHattwell/wormjam). Changes made throughout this publication are available in the master branch of the WormJam GitHub repository. Metabolites related to the different classes covered in this publication have been isolated based on their ID and name.

### 2.2. BiGG Metabolites

A text file containing all metabolites present in BiGG was downloaded (11/01/2020). An R script was used to isolate information on the individual DB identifiers. ChEBI, KEGG, HMDB, Lipidmaps and the InChIKey identifiers were used for further analysis [17,18,19,20,21]. Metabolites related to the different classes covered in this publication were isolated based on their ID and name.

### 2.3. WormJam Reaction and Metabolite Curation

Metabolic pathways and reactions were curated based on textbook knowledge and previously described gene associations in *C. elegans*. The new reactions do not contain any lumped or nested reactions. Several intermediate metabolites were newly generated. Both, neutral and charged structures were curated. Structures currently not available in different databases were drawn in ChemAxon Marvin Sketch and exported as InChI, InChIKey and SMILES. Identifiers in different databases were searched using BridgeDB and the bridgedbR package [22].

## 3. Results and Discussion

### 3.1. Evaluation of BiGG IDs and Naming for Acyl-Based Metabolites

According to the BiGG models ID specification and guidelines, IDs for metabolites should be human readable, short and memorable. However, when curating reactions manually it is also important to derive (at least partial) structural information from the ID to simplify the curation process. This is especially true for long reaction sequences, e.g., biosynthesis or β-oxidation of fatty acids. Inspecting IDs used in different BiGG models with different levels of structural details in the different IDs and naming were observed. IDs and naming of acyl-based metabolites, such as fatty acids, acyl-CoAs, acyl-carnitines and acyl-ACPs were downloaded from BiGG, and where available, chemical information was used. Interestingly, a large proportion of metabolites had no structural information associated with the BiGG DB or IDs from the chemical structure databases.

In order to evaluate if IDs and naming enable an unambiguous identification of the molecular entities within the different classes, they were grouped into different categories. The first group was called “full structural information” if structural information could be derived from the ID, name or links to unique chemical structures were available, “partial structural information” if minor information was missing, or “no structural information” if major information, e.g., position and/or stereochemistry of a double bond or functional group, was missing. An example of an ID and name pair that was classified as “full structural information” is lnlccoa, which is the ID for linoleyl-CoA. Ttdcrn, tetradecanoyl carnitine, was classified as “partial structural information” since the stereochemistry of the carnitine is missing, while ttdcea, tetradecanonate (n-C14:1) is classified as “no structural information”, because important information like the position and stereochemistry of the double bond is missing.

For fatty acids, acyl-CoAs and ACPs >66% had full structural information. Acyl-carnitines showed the highest number of partial structural information, mostly because the stereochemistry of the carnitine was missing.

Analysis of names, IDs and potentially linked chemical information revealed that for most metabolites, structural information can be deduced. Still, several metabolites with only partial or no structural information are present.

In order to explore if annotations of metabolites grouped as “no structural information” can be improved, the reactions in which these metabolites occur were checked. It might be possible that an upstream metabolite contained the full structural information, but this information was not propagated correctly. As an example, the metabolite ttdcea, annotated as tetradecenoate, was searched for in BiGG. This metabolite is used in many different models. The *Escherichia coli* model iJO1366 was selected, where this metabolite occurs in the cytosol and is linked to 9 reactions [23]. Selecting the reaction “FA141ACPHi” (Fatty-acyl-ACP hydrolase), it could be seen that ttdcea is derived from a more detailed tdeACP, which is described as cis-tetradec-7-enoyl-[acyl-carrier-protein] (n-C14:1). Using this information, ttdcea should be annotated as cis-7-tetradecenoate acid instead of tetradecenoate (n-C14:1). Cis-7-tetradecenoate can be found as metabolite M00117 in BiGG derived from Recon 3D [24].

Based on the idea that the reaction network can be used to check for connections between identifiers of different quality, reactions from BiGG were used to isolate pairs of acyl-based metabolites as they occur as substrate and product of reactions. All pairs that contained hub metabolites such as CoA, ccetyl-CoA, carnitine or ACP, as well as pairs derived from transport reactions, were removed. Metabolites were then labeled according to their group and connection of metabolites of different groups were checked. Table 1 summarizes these pairings.

Based on these pairings, it was determined that metabolites grouped into the category “full structural information” are mostly connected to metabolites of the same group. However, several connections between metabolites classified as “full structural information” and “no structural information” metabolites also exist. These connections together with known biochemistry can be used to improve annotation of metabolites. Furthermore, plotting all pairings as a network showed several long-distance possibilities for improvement. As result of this comparison, the annotation of specific metabolites can be improved. A particular example is the metabolite arachdcoa_c, which is annotated as C20:4-CoA. Although the identifier suggests that this metabolite might be arachidonoyl-CoA, no supporting information is supplied. Looking into connected metabolites, arachd_c was identified as a metabolite classified as “full structural information”. The name arachidonic acid identifies it with specific positions and stereochemistry of the double bonds (5Z,8Z,11Z,14Z), which identifies arachdcoa_c as arachidonyl-CoA. As a second reaction in links arachdcoa_c to adrncoa_c, adrenyl-CoA is also grouped as “full structural information”, which in turn is linked again to adrn_c, adrenic acid. This example shows how the network can be used to improve structural annotation of different metabolites to full structural detail.

### 3.2. Use of A Systematic Shorthand Notation to Generate IDs

Results so far show that the metabolic reactions and the obtained network can be used to improve the structural annotation of acyl-based metabolites. However, even IDs that have full structural details from different models are very different in the way they encode this structural information. All investigations so far have shown that a more systematic way of naming IDs of such acyl-based metabolites is required to avoid future confusion.

In lipidomics, the use of shorthand notations instead of long systematic IUPAC names is widespread to describe lipid structures. This shorthand notation has been standardized by Liebisch et al. and is the most widespread notation, being continuously improved to cover more lipid classes and modifications [15]. Such a systematic notation would resolve problems of cross-mapping between models; however it is not compliant with the specifications for BiGG metabolite identifiers. Transfer and adoption of this nomenclature for BiGG IDs will improve readability of IDs as well as structural details. A similar idea was already followed in the Chinese hamster ovary cell consensus model iCHOv1, where, for example, c81_5Zcrn_m was used to describe a carnitine molecule with a C8 chain and a cis-double bond at position 5, but no rules on how to generate the IDs or an automatic way have been proposed [25].

A typical example for a shorthand notation used in lipidomics to describe a fatty acid or acyl-CoA is shown in Figure 1. Structural features of an acyl chain are encoded in human readable abbreviations. First, the class of the molecule is denoted and in brackets structural features are encoded. Next, the number of carbons of the longest chain followed by the number of double bonds separated by a “:” is given. In a further pair of brackets, structural details like the position and geometry of double bonds or position and potential stereochemistry of functional groups are given.

At the current stage the following functional groups that are supported in the lipidomics shorthand notation: keto (O), hydroxy (OH), peroxy (OOH), amino (NH^2^) and methyl (Me) groups. If the groups have a stereocenters, they are defined directly after the functional group in square brackets, e.g., OH[S]. Each group additionally needs a number indicating the position for example 3OH[S]. Individual groups are separated by a comma. The order of function groups is double bonds, hydroxy groups, peroxy groups, keto groups, amino groups and methyl groups (DB > OH > OOH > O > NH_2_ > Me).

Similar rules are also applied to generate IDs for GSMs following the guidelines for valid BiGG IDs. First, the class of the molecule is denoted in lower case letter directly followed by the number of carbons and the double bonds separated by an underscore “_” (e.g., fa18_2). After a second underscore “_” the functional groups are followed as a combination of position and lower-case letters with no separator (e.g., fa18_2_9z12z).

If the functional group, e.g., a hydroxyl group, represents a stereocenter, the stereochemistry is directly indicated with the functional group by a capital letter (e.g., coa18_0_3ohS). Stereocenters of the base molecules, such as carnitine, are denoted similar to amino acids at the end with two underscores “__” (e.g., carn18_1_9z__L).

Nomenclatures for stereocenters shall not be mixed and only either R/S or D/L nomenclature is used within one identifier. Table 2 summarizes a few examples for the first round of β-oxidation of oleic acid.

If the shorthand notation of an acyl-based metabolite is known, it can be used to generate the corresponding identifier. A corresponding function has been added to the lipidomicsUtils R package (https://github.com/michaelwitting/lipidomicsUtils, unpublished). This function accepts a valid shorthand notation for different fatty acyls and creates the new identifier. For all metabolites with sufficient structural details, structures were curated from ChEBI or were drawn manually. The list of metabolites and their structures can be found in the Appendix A.

### 3.3. Comparison of Fatty Acid, Acyl-ACP and Acyl-CoA ID Naming in C. elegans GSMs

While working on the *C. elegans* consensus GSM, WormJam, it was realized that sections of the fatty acid biosynthesis, metabolism and β-oxidation pathways are poorly described and thus need further development. Fatty acid biosynthesis in *C. elegans* is well established based on the work of Watts and Browse [26]. Currently, a major curation aim for the WormJam team is the removal of lumped or nested reactions within the model, to allow a more detailed description and investigation of specific gene/enzyme reaction associations.

One example in the WormJam model was the biosynthesis of branched-chain fatty acids. *C. elegans* was shown to almost completely synthesize (>99%) its own branched-chain fatty pool [27]. Branched-chain fatty acids are important molecules and are required for the biosynthesis of C17iso sphingoid bases in *C. elegans* [28,29,30].

Searching for reactions associated with branched-chain fatty acids, several lumped/nested reactions have been identified, e.g., the following reaction, which produces the fatty acid 11-methyldodecanoic acid (C13:0iso) catalyzed by FASN-1.

11.0 M_h_c + 8.0 M_nadph_c + M_ivcoa_c + 4.0 M_malcoa_c

<=>

4.0 M_co2_c + 3.0 M_h2o_c + 5.0 M_coa_c + 8.0 M_nadp_c + M_fa13p0iso_c

This reaction is stoichiometrically correct, but misses several important aspects of fatty acid biosynthesis, e.g., the involvement of the Acyl-Carrier-Protein. In addition, the following reactions were found, describing the further elongation of C13:0iso to C15:0iso to C17:0iso:
3.0 M_h_c + 2.0 M_nadph_c + M_malcoa_c + M_fa13p0iso_c
<=>
M_co2_c + M_h2o_c + M_coa_c + 2.0 M_nadp_c + M_fa15p0iso_c
3.0 M_h_c + 2.0 M_nadph_c + M_malcoa_c + M_fa15p0iso_c
<=>
M_co2_c + M_h2o_c + M_coa_c + 2.0 M_nadp_c + M_fa17p0iso_c

Likewise, these reaction descriptions miss the important fact that the elongation reactions are carried on acyl-CoAs and not on the free fatty acids, as well as the involvement of ATP to generate an acyl-CoA from the free fatty acid. These reactions were inferred from iCEL1273, because ElegCyc did not contain any reactions related to branched-chain fatty acid biosynthesis. Several other errors have been found. In order to correct these errors, reactions were created from scratch using textbook knowledge and published gene associations. Searching for reactions and metabolite IDs in WormJam and BiGG, it was found that different classes of metabolites required for the biosynthesis and elongation as well as β-oxidation have different “namespaces” and use different abbreviations for the same acyl moiety. One particular example is summarized in Table 3.

While all IDs fulfill the requirement to be unique and short, they are hardly human readable because different abbreviations for the tetradecanoyl moiety are used (ttdc, td or ttd). This is also the case for other acyl-based metabolites, and the problem becomes more complicated if intermediates of biosynthesis and β-oxidation are included (trans-2-enoyl, 3-hydroxy and 3-keto derivates). Several reactions existed in duplicates due to differences in metabolite ID naming, which could be not be resolved automatically, especially as metabolite information, such as formulas or structures were often missing.

The metabolites in the WormJam version used here have already undergone several rounds of structural and naming curation. Therefore, most of the metabolites were scored to have full structural information. The majority of metabolites having only partial structural information were acyl-carnitines, because information on the stereochemistry of carnitine was missing. Because the WormJam model uses free L-carnitine, the stereocenter should be preserved in all reactions. Moreover, a large proportion of identifiers were not found in BiGG, ranging from 31% for fatty acids to 65% for acyl-carrier proteins. Particular examples are branched-chain fatty acids; even within the WormJam model, three different identifiers for 11-methyldodecanoic acid have been used: c13iso, C13iso and fa13p0iso. Results are summarized in Figure 2A.

### 3.4. Updating the WormJam Fatty Acid Related Reactions

Using the new suggested identifiers, reactions for the biosynthesis and peroxisomal as well as mitochondrial β-oxidation of fatty acids were curated. Fatty acid biosynthesis in *C. elegans* is performed by FASN-1, which produces palmitic acid or 11-methyldodecanoic acid, which are then further elongated and desaturated in the ER.

First, reactions producing palmitic and 11-methyldodecanoic acid were generated, which rely on ACP. Both cytosolic and mitochondrial versions were added to the model. In total, these two pathways added 110 new detailed reactions, which include each individual step of biosynthesis (condensation, 1st reduction, dehydration, 2nd reduction). In addition, 108 reactions for the elongation and desaturation of fatty acids in the ER were added at the same level of detail.

Beta-oxidation of fatty acids in the mitochondria and peroxisomes is used for the breakdown of fatty acids. In total, 124 reactions for the complete oxidation of saturated straight chain and branched-chain fatty acids were added. Thirty-nine reactions for the import of fatty acids into the mitochondria via the carnitine shuttle were added, as well as five additional reactions for fatty acid, which can enter by diffusion.

Lastly, oxidation of unsaturated fatty acids was added to the model. In total, the following reactions were added: β-oxidation of palmitoleic acid, β-oxidation of oleic acid, β-oxidation of vaccenic acid, and β-oxidation of linoleic acid. This added 49 additional reactions. At specific points, these reaction sequences feed into the sequences of β-oxidation of saturated fatty acids.

For illustration of the use of the new systematic identifiers, Figure 3 illustrates the synthesis of palmitoleic acid (Figure 3A), and the elongation of it to vaccenic acid, as well as the β-oxidation in the mitochondria (Figure 3B). All created reactions can be found in the Appendix A and can be used as a template for the generation of similar pathways.

## 4. Conclusions

Genome scale metabolic models (GSMs) are a powerful tool for investigations into metabolism. Metabolite mapping is an important part of using GSMs in combination with omics techniques. However, the annotation of compounds in the metabolic model is not always complete, and in most cases the annotation only contains the metabolite name and a formula. In particular, different acyl-based metabolites, such as fatty acids, acyl-CoAs and acyl-carnitines suffer from incomplete annotation, and missing position and stereochemistry information. Here a new and systematic naming for metabolite identifiers in GSMs for acyl-based metabolites is presented. Naming is based on a nomenclature used in the lipidomics field and follows certain specifications. Common nomenclature across different classes makes it possible to map between acyl-based metabolite classes. This new systematic nomenclature was used for the curation of fatty acid biosynthesis and degradation in the *C. elegans* consensus model.

Furthermore, the new nomenclature also makes it easy to introduce new classes of acyl-based metabolites. A particular example is N-acylethanolamides. BiGG contains several of these metabolites, mostly as part of Recon3D; oleth represent oleyl-ethanolamide, which could be represented as nae_18_1_8z with nae as an abbreviation for N-acylethanolamides.

Furthermore, the nomenclature can be further developed to also cover complex lipids. Different lipid classes can have up to four different acyl-groups (cardiolipins). Shorthand notations for lipids exists, but transfer of them to an identifier similar to the described ones would yield very long strings that would violate rules for valid BiGG identifiers.

## Figures and Tables

**Figure 1 metabolites-10-00130-f001:**
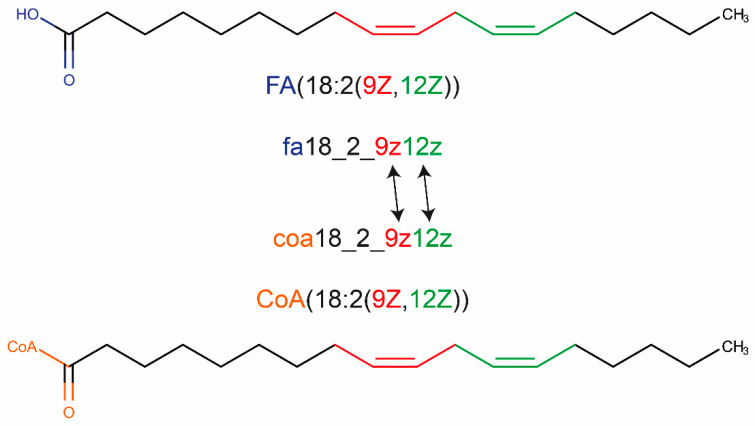
Example of structures, shorthand notation based on Liebisch et al. and the systematic nomenclature for acyl-based IDs.

**Figure 2 metabolites-10-00130-f002:**
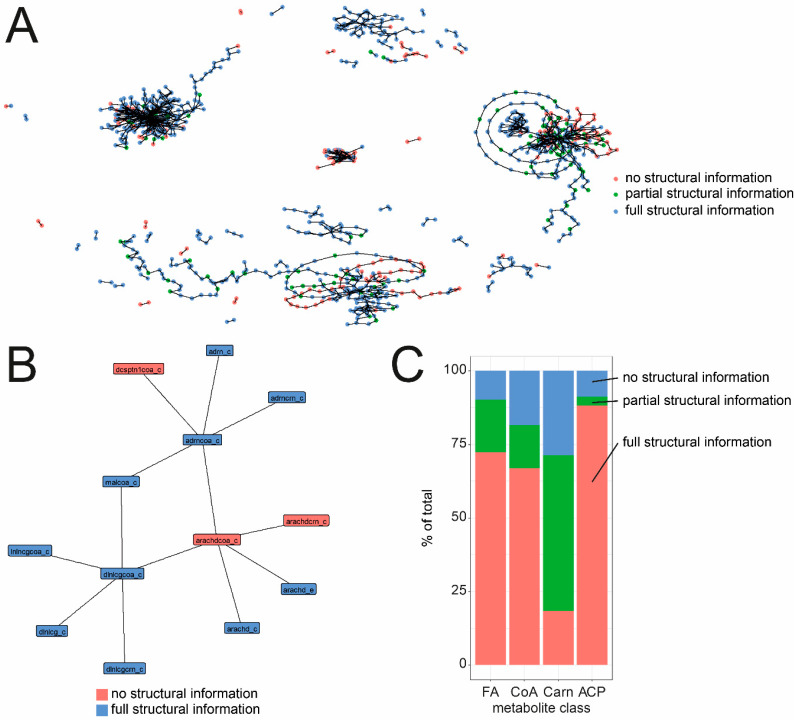
(**A**) Network of pairwise connections between metabolites classified into different structural details. Pairs were isolated from all reactions in BiGG and pairs containing hub metabolites were removed. (**B**) Subnetwork related to the metabolite arachdcoa_c indicating that this metabolite is connected to several metabolites classified to have full structural information. (**C**) Percentage of metabolites grouped into the different classes.

**Figure 3 metabolites-10-00130-f003:**
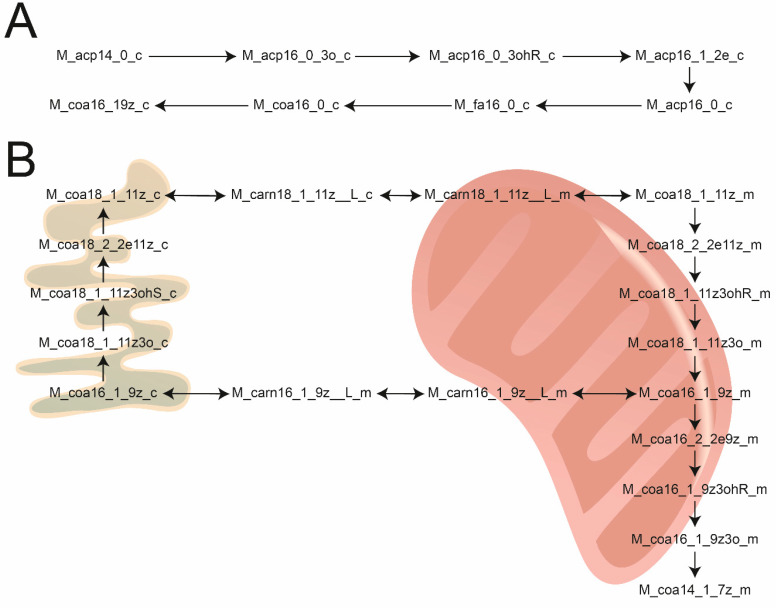
(**A**) Example reaction sequence for the biosynthesis of (9Z)-hexadecenoyl-CoA in *C. elegans*. (**B**) Example reaction sequences for the elongation of fatty acids, carnitine shuttle and β-oxidation in the mitochondria. For all examples, the new systematic identifiers were used, which enabled human readable curation of reactions.

**Table 1 metabolites-10-00130-t001:** Count of pairings between metabolites classified with the different structural details.

Pairing	Count
no structural information<-> no structural information	135
partial structural information <-> no structural information	8
Partial structural information <-> partial structural information	36
Full structural information <-> no structural information	96
Full structural information <-> partial structural information	161
Full structural information <-> full structural information	897

**Table 2 metabolites-10-00130-t002:** Comparison of reaction sequence from the first step of beta oxidation of oleic acid.

Name	Shorthand Notation	Old	New
Oleyl-l-carnitine	Carn(18:1(9Z))	ocdce9crn	carn18_1_9z__L
Oleyl-CoA	CoA(18:1(9Z))	odecoa	coa18_1_9z
(2E,9Z-Octadecenoyl-CoA	CoA(18:2(2E,9Z))	od29coa	coa18_2_2e9z
(S)-3-Hydroxyl-Oleoyl-CoA	CoA(18:1(9Z,3OH[S]))	3hod9coa	coa18_1_9z3oh__S
3-Oxo-Oleoyl-coA	CoA(18:1(9Z,3O)	3ood9coa	coa18_1_9z3o
(7Z)-Hexadecenoyl-CoA	CoA(16:1(7Z))	hd7coa	coa16_1_7z

**Table 3 metabolites-10-00130-t003:** Comparison of names for different tetradecanoic acid derived metabolites.

	iCel1273	ElegCyc	WormJam	BiGG	New ID
Tetradecanoic acid (Myristic acid)	ttdca	ttdca	ttdca	ttdca	fa14_0
Tetradecanoyl-Acyl-Carrier Protein (Myristoyl-ACP)	Myristoyl_ACPs	---	myrsACP	myrsACP	acp14_0
Tetradecanoyl-Coenzyme A (Myristoyl-CoA)	tdcoa	tdcoa	tdcoa	tdcoa	coa14_0
Tetradecanoyl-Carnitine (Myristoyl-Carnitine)	ttdcrn	CPD909_16	ttdcrn	Ttdcrn M02973	crn14_0__L

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
