# Peer review of "Suggestions for Standardized Identifiers for Fatty Acyl Compounds in Genome Scale Metabolic Models and Their Application to the WormJam Caenorhabditis elegans Model"

_metabolites, 2020, doi:10.3390/metabo10040130_

Round 1
Reviewer 1 Report
The manuscript deals with the important issue of standardization of nomenclature in genome-scale metabolic models (GSMs). It proposes a new nomenclature for acyl-based metabolites within the GSMs to directly encode the structural details in the metabolite identifiers. Its application is demonstrated on fatty acid biosynthesis and degradation within the WormJam GSM. The significance of the manuscript could be increased with the proposal of the methodology for the automatic generation of metabolite identifiers in the proposed format.
Major comments:
How much additional work is required to deduce the identifiers in the process of GSM reconstruction? The adoption of this standard within the GSM community is questionable if a significant amount of additional manual work and curation is required.
Why only acyl-based metabolites? How much effort would be required to apply the proposed nomenclature to all the metabolites within the GSM?
Please elaborate on the possibilities of the automatic generation of the metabolite identifiers in the proposed format. The significance of the manuscript would be increased significantly if a methodology for an automatic generation would be proposed, implemented and integrated within constraint-based reconstruction and modeling frameworks, such as COBRA.
Minor comments:
Line 263: "Table 3" instead of "Table 1"
Line 273: "grade" is missing.
Author Response
Reviewer 1
The manuscript deals with the important issue of standardization of nomenclature in genome-scale metabolic models (GSMs). It proposes a new nomenclature for acyl-based metabolites within the GSMs to directly encode the structural details in the metabolite identifiers. Its application is demonstrated on fatty acid biosynthesis and degradation within the WormJam GSM. The significance of the manuscript could be increased with the proposal of the methodology for the automatic generation of metabolite identifiers in the proposed format.
Major comments:
How much additional work is required to deduce the identifiers in the process of GSM reconstruction? The adoption of this standard within the GSM community is questionable if a significant amount of additional manual work and curation is required.
The identifier can be generated from the chemical name or the lipid shorthand notation according to Liebisch et al, if available. For all elaborated acyl-based metabolites used and for which enough structural detail was available the new IDs have been generated, as well as structures have now been curated from ChEBI or added manually. If certain metabolites and metabolic pathways are refactored the IDs from this publication can be used.
Why only acyl-based metabolites? How much effort would be required to apply the proposed nomenclature to all the metabolites within the GSM?
The proposed nomenclature is tailored towards acyl-based metabolites and is based on the shorthand notation by Liebisch et al. During the curation of reactions from the WormJam model it was realized that acyl-based metabolites have generally only weak IDs.
For other metabolites nomenclature based on trivial names might be more useful and human readable, e.g. ala__L clearly identifies L-Alanine. Current efforts towards standardization are bundled by new tools such as the recently published Memote (https://www.nature.com/articles/s41587-020-0446-y). Use of common identifiers in the models is a common topic that is currently followed by different research groups.
Please elaborate on the possibilities of the automatic generation of the metabolite identifiers in the proposed format. The significance of the manuscript would be increased significantly if a methodology for an automatic generation would be proposed, implemented and integrated within constraint-based reconstruction and modeling frameworks, such as COBRA.
If the shorthand nomenclature for the acyl-based metabolites according to Liebisch et al. exists, the new ID can be generated from this nomenclature. A corresponding function has been added to the unpublished lipidomicsUtils R package (https://github.com/michaelwitting/lipidomicsUtils).
Integration into model generation can be achieved once the nomenclature has been taken up by the community. During generation of model’s reactions are typically fetched from reaction databases, if the new nomenclature is present it can be directly used.
In order to further foster the use of the new metabolite identifiers all curated WormJam reactions in which the new identifiers have been used were added as additional supplementary information. These reactions can be used as template for constructions of other models.
Minor comments:
Line 263: "Table 3" instead of "Table 1"
Line 273: "grade" is missing.
Thank you very much for catching. The errors were corrected accordingly.
Reviewer 2 Report
Witting et al present a new, unified systematic nomenclature for acyl-based metabolites such as fatty acids, acyl-carnitines, etc, with a focus on C. elegans. Such a system should enable easier use across various systems and enabled comparing of data more easily. A key focus is that the new nomenclature is similarly readable and understandable for humans and computer systems alike.
The work is informative and useful; as always, its ultimate utility will depend on how much people will use it which is impossible to predict. The work is well done, organized, and presented, and the figures are accessible and understandable.
I have only a few minor typo comments:
- L40: use plural either “formulas” or “formulae”
- L50: use past tense “led” instead of “lead”
- L57: “other” should be plural
- L61: “pathway” should be plural
- L68: "is often only hardly possible” —> "is often difficult”
- L88: replace “goodness” with validity or similar
Author Response
Reviewer 2
Witting et al present a new, unified systematic nomenclature for acyl-based metabolites such as fatty acids, acyl-carnitines, etc, with a focus on C. elegans. Such a system should enable easier use across various systems and enabled comparing of data more easily. A key focus is that the new nomenclature is similarly readable and understandable for humans and computer systems alike.
The work is informative and useful; as always, its ultimate utility will depend on how much people will use it which is impossible to predict. The work is well done, organized, and presented, and the figures are accessible and understandable.
I have only a few minor typo comments:
- L40: use plural either “formulas” or “formulae”
- L50: use past tense “led” instead of “lead”
- L57: “other” should be plural
- L61: “pathway” should be plural
- L68: "is often only hardly possible” —> "is often difficult”
- L88: replace “goodness” with validity or similar
Thank you very much for the very positive comments. We corrected all typos and performed another round of grammar and style correction.
Round 2
Reviewer 1 Report
The author has adequately addressed my comments.